# Fast Synthesis of Au Nanoparticles on Metal–Phenolic Network for Sweat SERS Analysis

**DOI:** 10.3390/nano12172977

**Published:** 2022-08-28

**Authors:** Xiaoying Zhang, Xin Wang, Mengling Ning, Peng Wang, Wen Wang, Xiaozhou Zhang, Zhiming Liu, Yanjiao Zhang, Shaoxin Li

**Affiliations:** 1Department of Physical Education, Guangdong Medical University, Dongguan 523808, China; 2School of Medical Technology, Guangdong Medical University, Dongguan 523808, China; 3Guangzhou Key Laboratory of Spectral Analysis and Functional Probes, College of Biophotonics, South China Normal University, Guangzhou 510631, China; 4School of Basic Medicine, Guangdong Medical University, Dongguan 523808, China; 5School of Biomedical Engineering, Guangdong Medical University, Dongguan 523808, China

**Keywords:** surface-enhanced Raman scattering, metal–phenolic network, sweat, molecular fingerprint, pH

## Abstract

The biochemical composition of sweat is closely related to the human physiological state, which provides a favorable window for the monitoring of human health status, especially for the athlete. Herein, an ultra-simple strategy based on the surface-enhanced Raman scattering (SERS) technique for sweat analysis is established. Metal–phenolic network (MPN), an outstanding organic-inorganic hybrid material, is adopted as the reductant and platform for the in situ formation of Au-MPN, which displays excellent SERS activity with the limit of detection to 10^−15^ M for 4-mercaptobenzoic acid (4-MBA). As an ultrasensitive SERS sensor, Au-MPN is capable of discriminating the molecular fingerprints of sweat components acquired from a volunteer after exercise, such as urea, uric acid, lactic acid, and amino acid. For pH sensing, Au-MPN/4-MBA efficiently presents the pH values of the volunteer’s sweat, which can indicate the electrolyte metabolism during exercise. This MPN-based SERS sensing strategy unlocks a new route for the real-time physiological monitoring of human health.

## 1. Introduction

Sweat is closely related to the physiological state of the human body, and its composition and pH value reflect different health conditions of the human body. Sweating has not only the function of regulating body temperature but also is an important excretory pathway to maintain a normal metabolism [1,2]. Sweat often contains a variety of electrolytes and metabolites, such as urea, uric acid, and lactic acid, as well as toxic substances after medication [3,4]. In view of the accessibility of sweat, the effective monitoring of sweat components is expected to be a simple and real-time means to assess human health status [5,6,7]. Exercise can accelerate the discharge of sweat, accompanied by a large number of metabolites, neurotransmitters, and other chemical molecules [8,9]. Ions such as Na^+^, Cl^−^, and K^+^ can also be flown out with perspiration, thereby affecting the pH value of sweat [10,11]. Excessive exercise may lead to excessive loss of mineral components in the body, causing electrolyte imbalance and endangering health [12]. The use of real-time portable photoelectric sensors helps to provide athletes with scientific sports data, formulate better training programs, and avoid training risk [13]. However, the accurate detection of a tiny volume of sweat is still a challenge [14].

The performance of photoelectric sensors largely depends on the properties of materials. Metal–phenolic network (MPN), an outstanding organic-inorganic hybrid material firstly reported by Caruso’s group [15], has been widely used as a versatile coating material for multiple applications [16,17,18,19,20]. Different from other supramolecular structures, the construction cost of MPN is low, and its synthesis is extremely simple that can be assembled on a large scale within 20 s [21]. For photoelectric sensing applications, MPN can be used as an excellent adhesive material for loading fluorescent molecules to realize multicolor fluorescent labeling and biosensor [22]. MPN can also serve as the soft metal probe for ultrasensitive electrochemical immunoassay [23]. In addition, MPN exhibits excellent reducing capacity allowing the fast generation of metal nanoparticles (NPs) on arbitrary material, which displays enhanced photoelectric features for photocatalysis and anti-microorganism [24,25,26]. Owing to the high affinity of MPN to biomolecules, the visualization and spectroscopic analysis of latent fingerprints have been realized by the formation of metal NPs on MPN [27]. This strategy is recently adopted for the fabrication of a three-dimensional surface-enhanced Raman scattering (SERS) sensor by in situ reductions of Ag NPs on MPN-coated commercial nanoanodic aluminum oxide film, achieving an ultrasensitive detection of raticide [28]. SERS has been proved to be a powerful analytical tool capable of providing the molecular fingerprint of the target [29]. SERS provides availability for the identification of biochemical components of a tiny sample with single-molecule accuracy [30,31]. SERS monitoring of the biochemical components in sweat samples is appealing, which can supply indicators of human health [32,33,34]. SERS-active nanostructures are also combined with flexible materials to fabricate wearable SERS sensors for real-time sweat monitoring [35,36,37]. Noble metal NPs such as Au, Ag, and Cu are commonly used as the active substrates for SERS analysis due to the Raman electromagnetic enhancement caused by the localized surface plasmon resonance (LSPR) effect. However, the SERS activities of discrete metal NPs are limited by their sparse “hot spots”.

Herein, a simple strategy for the SERS sensor is proposed by in situ rapid generations of Au NPs on MPN (Figure 1). The as-prepared Au-MPN displays a film-like structure with densely scattered Au NPs, which generates abundant “hot spots”, achieving excellent SERS activity with the limit of detection (LOD) to 10^−15^ M for 4-mercaptobenzoic acid (4-MBA). The general surface binding affinity of MPN shows the great convenience of the nanocomposites to fabricate versatile SERS platforms. Au-MPN is then used as an ultrasensitive SERS substrate for the detection of the biochemical components of sweat samples acquired from a volunteer after exercise. In addition, a pH nanoprobe based on Au-MPN is established to monitor the electrolyte metabolism of the volunteer.

## 2. Materials and Methods

### 2.1. Reagents

Tannic acid (TA), chloroauric acid (HAuCl_4_·4H_2_O), and polyvinylpyrrolidone (PVP, K30) were purchased from China National Medicine Corporation (Shanghai, China). 3-(N-morpholino)propanesulfonic acid (MOPS) was obtained from Aladdin Reagent Co. Ltd. (Shanghai, China). Simulated sweat (pH 4.7, containing NaCl, urea, fatty acid, lactic acid, etc.) was purchased from Nanjing Xinfan Co. LTD, China. FeCl_3_·6H_2_O, 4-MBA, and other reagents were used as analytically pure grade. Deionized water (Milli-Q System, Millipore, Billerica, MA, USA) was used throughout the experiments.

### 2.2. Synthesis of MPN

PVP (1.5 mg/mL) and FeCl_3_·6H_2_O (0.12 mg/mL) were added to a flask containing deionized water (10 mL). Then TA (0.4 mg/mL) was added into the mixture and stirred for 20 min at room temperature. The as-prepared MPN was collected by centrifugation (3000× *g*, 10 min), and resuspended in deionized water.

### 2.3. Preparation of Au-MPN

HAuCl_4_ (0.4 mM) was added into the eppendorf tubes containing 0.1 mg/mL MPN solution and quickly placed on a vortex mixer for 2 min. After standing for 10 min, the mixture was centrifuged at 3000 rpm for 10 min to collect Au-MPN. For the preparation of Au-MPNs with different Au/MPN ratios, HAuCl_4_ with different concentrations (0.1, 0.2, 0.3, 0.4 and 0.5 mM) were used, named Au-MPN-1, Au-MPN-2, Au-MPN-3, Au-MPN-4 and Au-MPN-5, respectively.

### 2.4. Preparation of pH SERS Probe

4-MBA (10^−4^ M) was added into the eppendorf tubes containing Au-MPNs with various Au/MPN ratios and quickly placed on a vortex mixer for 2 min. After standing for 10 min, the mixture was centrifuged at 3000 rpm for 10 min to collect the pH-responsive SERS nanoprobe (Au-MPN/4-MBA).

### 2.5. Characterization

Transmission electron microscopy (TEM) was performed using a JEM-2100HR transmission electron microscope (JEOL, Tokyo, Japan) operated at 200 kV. Ultraviolet-visible (UV-VIS) absorbance spectra were acquired using a UV-VIS spectrometer (UV-6100, MAPADA, Shanghai, China). Raman spectra were recorded using a confocal microspectrometer (InVia, Renishaw, Derbyshire, England) equipped with a 785 nm laser. Fourier transform infrared (FT-IR) spectra were taken using a Nicoleti S50 spectrometer (Thermo, Waltham, MA, USA).

### 2.6. SERS Experiments

In order to optimize the Au/MPN ratio, Au-MPNs after grafting of 4-MBA were diluted into 0.2 mM and placed under the Renishaw Raman microscope for detection. A 20× objective lens was used to focus the laser beam and to collect the Raman signal. The Raman spectrum was acquired at a center wavelength of 1200 cm^−1^ under the 785 nm laser irradiation (5 mW, 6 s accumulation time). All the samples were measured at least five times.

Then, the optimal Au-MPN was mixed with different concentrations of 4-MBA (10^−4^–10^−15^ M) and detected under the Raman spectrometer to study the LOD of Au-MPN. For SERS repeatability analysis, the pH SERS probe was dropped onto an Al wafer to dry, and the SERS spectra at random 40 spots were collected. The intensity of the Raman band at 1078 cm^−1^ was used to calculate the relative standard deviation (RSD) value. For SERS image, the Raman mapping was performed in the streamline mode at wavenumber center 1200 cm^−1^ with 1 s of laser exposure (785 nm, 10 mW). The lateral resolution was 28 μm, and a total of 3658 spectral lines were collected.

### 2.7. SERS Fingerprint Analysis of Sweat

Eleven volunteers (8 men, 3 women) with an age range of between 22–36 were recruited, and the volunteers gave written consent to participate in the study, which was approved by the Ethics Committee of Guangdong Medical University. After exercise, the sweat samples of the volunteers were collected for further analysis. Then 1 μL of sweat was dropped onto the pre-deposited Au-MPN substrate for SERS detection (785 nm laser, 10 mW, 6 s exposure time).

### 2.8. SERS pH Detection of Sweat

MOPS buffer solutions with the pH value from 3 to 10 were first prepared to assess the pH response characteristics of Au-MPN/4-MBA. One μL of pH sensor (2 mM) was mixed with 10 μL MOPS buffer, and the SERS spectrum was acquired. The pH-related SERS bands were selected to plot the pH response curve. After that, the simulated sweats with different pH values were used to record the pH-responsive dynamics of Au-MPN/4-MBA in order to get rid of the interference of other components in sweat. The pH value of real sweat collected from volunteer was determined using Au-MPN/4-MBA as the pH sensor according to the same SERS protocol. All the SERS detection was repeated at least five times.

## 3. Results and Discussion

### 3.1. Characterization of the Nanostructures

The formation of MPN film commonly involves a process of solid-phase template growth and etching [15]. Recently, Chen et al. proposed a strategy for controllable self-polymerization of MPN in the liquid phase by using polymer molecules [16]. Here we used PVP as the liquid-phase template to support the self-assembly of MPN. The morphology of MPN detected by TEM is shown in Figure 1A, where a film-like and porous structure is observed. Figure 1B displays the microstructure of MPN after in situ reductions of Au NPs, in which spherical Au NPs with a mean size of 52 nm are randomly dispersed on MPN film. The high-resolution TEM (HRTEM) image of Au-MPN reveals the amorphous structure of MPN and a crystalline structure of Au NP with a lattice spacing of about 0.24 nm (Figure 1C). The UV-VIS absorbance spectrum of MPN illustrates a typical absorption band at around 570 nm, which is consistent with that reported previously [16]. After Au NPs generation, a strong LSPR peak at 540 nm emerges (Figure 1D), and the synthetic efficiency of Au-MPN is ideal (Appendix A). The Raman spectra of MPN and Au-MPN (Figure 1E) show the characteristic Raman bands at 1350 and 1480 cm^−1^, ascribing to the skeletal vibrations of the benzene ring [20]. In addition, the chemical composition of Au-MPN shows the common features of its precursors, as confirmed by the FT-IR spectra (Figure 1F), such as 1645 cm^−1^ (C=O stretching), 1440 cm^−1^ (C–O stretching), 1282 cm^−1^ (C–N stretching), 1192 cm^−1^ (O–H bending), and 1060 cm^−1^ (C–OH stretching) [26,38,39]. We also observe a reduced intensity of the C–OH band in the spectral lines of MPN and Au-MPN compared with that of TA, indicating the coordination of the phenolic groups with Fe ions [21].

### 3.2. SERS Activities of Au-MPN

In order to investigate the optimal SERS activity, five Au-MPNs with different Au/MPN ratios were prepared. A red shift of the LSPR peak is observed in the UV-VIS absorbance spectra of Au-MPNs (Figure 2A), indicating an increasing size of Au NPs on MPN (Appendix A). Then 4-MBA molecules were grafted onto Au-MPNs (Appendix A). Figure 2B displays the SERS spectra of 4-MBA induced by those Au-MPNs, where Au-MPN-4 triggers the strongest SERS signals that can well match with the fingerprints of 4-MBA (Appendix A), and Au-MPN-4 was used for further experiments.

Then, different concentrations of 4-MBA molecules were deposited onto Au-MPN-4 for SERS detection. As shown in Figure 3A, the concentration-dependent SERS signals are observed at the concentration range of 4-MBA from 10^−15^ to 10^−4^ M. The SERS spectral pattern retains well though some peaks disappear at low 4-MBA concentrations. The major bands at 1078 and 1596 cm^−1^ assigned to the ring-breathing modes always exist, even if the concentration of 4-MBA decreases down to 10^−15^ M. To estimate the SERS repeatability, more than 40 SERS spectral lines of 4-MBA after grafting onto Au-MPN were collected (Figure 3C); and the SERS intensity of the Raman band at 1078 cm^−1^ was shown in Figure 3B, in which the RSD value was calculated to be 2.1%, indicating a good SERS reproducibility. We also plotted a SERS map of Au-MPN/4-MBA after drying, further confirming the SERS’ good uniformity (Figure 3D).

### 3.3. Molecular Fingerprint of Human Sweat Induced by Au-MPN

The excellent SERS performance of Au-MPN endows it with great potential for the molecular fingerprint analysis of analytes with low scattering cross sections, such as biological samples. We recruited 11 volunteers (8 men, 3 women) and collected their sweats after exercise. The SERS spectra of these sweat samples were then acquired using Au-MPN as the ultrasensitive optical sensor. Figure 4A illustrates the molecular fingerprints of sweats of these 11 volunteers, where abundant fingerprint information is observed, such as 651, 736, 830, 856, 927, 1003, 1045, 1080, 1249, 1322, 1358, 1457, 1493, 1589, 1621 and 1648 cm^−1^, etc. The biochemical assignments of these Raman bands are listed in Table 1. It can be clearly noticed that many vibrational modes are assigned to the major components of sweat (urea, uric acid, lactic acid, and amino acid) [40,41,42,43,44], indicating the potent potential of the Au-MPN-based SERS technique for sweat identification. The SERS spectra of volunteer sweats share a basic spectral pattern; however, the obvious difference is still noticed among these 11 spectral lines. For example, most SERS spectra exhibit a strong Raman peak at 1003 cm^−1^, while the intensity of this band decreases dramatically in 5#, 9#, and 11#, revealing a low urea excretion in these three volunteers. In addition, a sharp SERS peak at 1033 cm^−1^ that can be ascribed to uric acid [41,44] is observed in the spectral line of 9#, which may be an indicator of some physiological changes.

We also monitored the sweat composition change during exercise using the Au-MPN-based SERS technique. Figure 4B shows the SERS spectra of sweat samples collected from volunteer 1# at an interval of 30 min after exercise. It can be observed that the integrated SERS intensities of sweat decrease gradually with the extension of exercise time, indicating the content of biochemical components in sweat reduces after long-term exercise. Moreover, we can also notice the fluctuation of fingerprint information in the SERS spectra, such as 736, 830, 927, 1045, 1457, 1493, and 1648 cm^−1^. Therefore, the Au-MPN-based SERS technique offers a simple and reliable sensing method for monitoring the physiological state of the human body.

### 3.4. pH Sensing Based on Au-MPN

The electrolyte content determines the pH value of sweat. 4-MBA is a well-defined pH-responsive molecule that has been widely used for pH sensing [14,45,46]. We also fabricated a pH SERS nanoprobe by covalently grafting 4-MBA onto Au-MPN. The thiol group of the 4-MBA molecule shows a strong affinity for Au NPs and forms a stable Au-S bond [47]. Thus, the Raman fingerprints of sulfur-contained molecules are more prone to be enhanced by the metal-based SERS substrates [48]. The SERS performance of Au-MPN/4-MBA in MOPS buffer solutions at different pH values is displayed in Figure 5A. It can be noticed that the SERS signals in three regions, 665–745 cm^−1^ (Figure 5B), 775–880 cm^−1^ (Figure 5D), and 1340–1460 cm^−1^ (Figure 5F), change significantly related to pH. For quantitative analysis, the intensity ratios of 722 to 698 cm^−1^, 845 to 805 cm^−1^, and 1400 to 1596 cm^−1^ as a function of pH were plotted, as shown in Figure 5C,E,G, respectively. The R^2^ values of the fitting curves of I_722_/I_698_, I_845_/I_805_, and I_1400_/I_1596_ are calculated to be 0.998, 0.994, and 0.989, respectively, indicating the exponential relationship between the SERS signals with pH.

Then, we used simulated sweat to evaluate the availability of Au-MPN-based pH sensors in the status close to a real environment. The SERS spectra (Figure 6A) and enlarged spectral lines in the ranges of 665–745 (Figure 6B), 775–880 (Figure 6D), and 1340–1460 cm^−1^ (Figure 6F) at various pH values exhibit similar trends in comparison with that in MOPS buffer. The fitting curves of I_722_/I_698_ (Figure 6C), I_845_/I_805_ (Figure 6E), and I_1400_/I_1596_ (Figure 6G) are also plotted, and the corresponding R^2^ values are calculated to be 0.999, 0.989, and 0.988, respectively. Among that, I_722_/I_698_ may be the best indicator for the pH of sweat. It can be concluded that Au-MPN/4-MBA displays promising potential as an efficient SERS nanoprobe for pH sensing of real sweat samples.

### 3.5. pH Monitoring of Human Sweat after Exercise

We finally utilized Au-MPN/4-MBA as novel pH SERS sensor for the detection of human sweat collected from a volunteer after exercise. The SERS spectra of 4-MBA in the sweat samples of the 11 volunteers are displayed in Figure 7A, where the Raman signals in the range of 665–745 cm^−1^ reflect the sweat pH. The pH values of the sweat samples were then calculated according to the fitting curve in Figure 6C. As demonstrated in Figure 7B, the pH values are situated in the range of 4.0–5.3 for the 11 volunteers. It can also be observed that the pH values of sweats for women are obviously higher than that for men. However, more sample data are needed to support this result. In addition, we monitored the change in pH value of sweat during exercise using Au-MPN/4-MBA. Figure 7C exhibits the SERS spectra and corresponding pH values of sweat samples acquired from volunteer 1# after exercise. We can notice that the pH value of sweat increases slightly during 1.5 h of perspiration, which may reflect the electrolyte metabolism level during exercise.

## 4. Conclusions

In conclusion, an ultra-simple strategy for fabrication of SERS substrate is proposed by in situ reductions of AuNPs on supramolecular metal–organic networks. The versatile binding affinity of MPN endows Au-MPN with excellent SERS activity and repeatability for molecular detection, which exhibits a LOD value of 10^−15^ M for 4-MBA. For sweat fingerprint analysis, the Au-MPN-based SERS technique can efficiently distinguish the biochemical composition of sweat, such as urea, uric acid, lactic acid, and amino acid, which offers a reliable sensing method for the monitoring of physiological state during exercise. Au-MPN is also used to construct a pH SERS sensor by covalently grafting a pH-responsive 4-MBA molecule. The exponential relationship between the Raman signals with the pH values of sweat is observed in SERS spectra of Au-MPN/4-MBA. Using it as a novel pH SERS nanosensor, the pH values of the sweat samples acquired from volunteers are calculated to be 4.0–5.3, reflecting the electrolyte metabolism level during exercise. This MPN-based SERS sensing strategy paves a new way for the real-time monitoring of exercise physiology.

## Data Availability

Not applicable.

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
