# Peer review of "Fast Synthesis of Au Nanoparticles on Metal–Phenolic Network for Sweat SERS Analysis"

_nanomaterials, 2022, doi:10.3390/nano12172977_

Round 1
Reviewer 1 Report
The manuscript entitled “Fast Synthesis of Au Nanoparticles on Metal-Phenolic Network for Sweat SERS Analysis” by Zhang et al. demonstrates the fabrication of Au-MPN to detect biomarkers in sweat. I liked very much of the pH sensing study; however, I have some doubts about the molecular fingerprint studies. How these measurements were performed must be clarified. Was the SERS substrate in liquid form, deposited on an Aluminium wafer or filter paper?? If the particles are deposited on a filter paper, where is the characterization of the substrates?
I think there are open questions that must be answered. So, I do not recommend this paper for publication.
See comments below:
Major questions:
1. Can the authors explain the role of the FeCl3 on the synthesis of Au-MPN particles. Is it still coordinating the tannic acid?
2.The authors claimed “For fabrication of flexible SERS substrate, a filter paper was immersed into the Au-MPN (10 mM) solution for 1 h, obtaining Au-MPN paper.” Where is the characterization of this substrate? SEM images , UV/VIS spectrum and digital images must be added to the manuscript.
3. The ratios used in section 2.4 must be described.
4. Raman mapping parameters must be added (area scan, number of Raman spectra, laser power, etc). Why is the band at 1200 cm-1 monitored if 4-MBA does not have any Raman band in that wavenumber?
5. The results must be discussed. For example, “In addition, the chemical composition of Au-MPN shows the common features of its precursors, as confirmed by the FT-IR spectrum (Figure 1F).” The FT-IR should be explained, and the vibrational modes of the precursors must be highlighted in the Au-MPN particles.
6. The authors claimed “A red-shift of the LSPR peak is observed in the UV-Vis absorbance spectra of Au-MPNs (Figure 2A), indicating the change in the size of Au-MPNs.” First, we do not know the ratios used to prepare the particles. Second, what are the particle sizes with the difference between ratios?
7. Figure 2B, why it is not observed the Raman features of Au-MPN for Au-MPN-1 like it is observed in Figure 1E? The conventional Raman of 4-MBA and molecular structure should be added to Figure 2B for comparison.
8. What particles did the authors use to perform the measurements presented in Figures 3C and D. This should be described in the caption.
9. How is the SPR of the Au particles after the functionalization with 4-MBA?
10. I do not believe that the authors observed Raman signal of 4-MBA at 10-15M. That particular Raman spectrum presented in Figure 3A is very similar to the one presented in Figure 1E. So, I believe that the authors are observing the Raman features of MPN particles. This must be revised.
11. The authors claimed, “We also plotted a SERS map of Au-MPN/4-MBA after drying, further confirming the SERS good uniformity (Figure 3D).” The particles were drying in what? Al wafer, filter paper? This must be very clear in the manuscript.
If the authors have deposited the Au-MPN particles in Al wafer, did they have the same detection limit using Silicon wafers or filter paper as supports? It is known that the Al wafer can enhance the Raman signal of molecules that are adsorbed in plasmonic particles.
12. Did the authors perform the conventional Raman spectra of the sweat samples? Can the authors demonstrate the result or add it to the supporting information. Can the authors confirm the SERS results (the presence of the biomarkers) in sweat using other techniques to confirm the feasibility of the study?
13. The authors claimed, “In addition, a sharp SERS peak at 1033 cm-1 that can be ascribed to uric acid”. If this band is from uric acid, why is not observed the bands at 736 and 1358 cm-1?
14. The authors claimed, “It can be also ob- served that the pH values of sweats for women are obviously higher than that for men”. How we can know if the sweat samples are anonymous.
15. The authors claimed, “We can notice that the pH value of sweat increases slightly during 1.5 hours of perspiration, which reflects the electrolyte metabolism level during exercise.”. I did not understand. What means, in sports, the increase of the pH during exercise?
Minor observations:
-2.1 reagents: FeCl3 and 4-MBA must be described as the other reagents. The components and pH of the simulated sweat must be described.
-It is UV/VIS instead of UV/Vis
-all the data with light red and light blue cannot be observed in the graphs (Figure 3C, Figure 4A, Figure 5A, etc)
Author Response
Dear Reviewer:
Thanks for your comments about our manuscript ID nanomaterials-1835650 entitled "Fast Synthesis of Au Nanoparticles on Metal-Phenolic Network for Sweat SERS Analysis" has been carefully revised according to Reviewers’ suggestions. And we hope to have the opportunity to Nanomaterials.
We have revised the manuscript accordingly and the revised portion is marked in red. We hope these will make it more acceptable for publication.
Yours sincerely,
Xiaoying Zhang, Xin Wang, Mengling Ning, Peng Wang, Wen Wang, Xiaozhou Zhang, Zhiming Liu, Yanjiao Zhang, Shaoxin Li
Correspondence author: Dr. Zhiming Liu
Guangzhou Key Laboratory of Spectral Analysis and Functional Probes,
College of Biophotonics,
South China Normal University,
Guangzhou 510631, China
E-mail: liuzm021@126.com

Reviewer 2 Report
In the manuscript entitled “Fast Synthesis of Au Nanoparticles on Metal-Phenolic Network for Sweat SERS Analysis” the Authors have described the fabrication of SERS substrate as proposed by in situ reductions of AuNPs on supramolecular metal-organic network and analysis of sweat fingerprint analysis.
This paper should be revised according to the following comments.
- In the introduction, part Authors should additional information regarding the Plasmon nature of AuNPs.
- In data interpretation please provide a more extended discussion about the selectivity of AuNPs – Sulfur-contain components in relation to SERS effects.
- What about the efficiency of synthesis? How the AuNPs after synthesis was clean up for unreacted Au ions? Please, discuss this issue and re-analyze the UV spectra.
In my opinion, the manuscript required a minor revision before final acceptance.
Author Response

(The authors gave the same response as above.)

Reviewer 3 Report
This is an interesting paper reporting a smart and innoviatve method for the analysis of sweat components. Authors offer good evidence that the proposed analytical metohd is working. The paper is bascially well doen adn well writtne. Some minor stylistic mistakes need to be corrected
Author Response

(The authors gave the same response as above.)
